# Isocitrate Dehydrogenase 1/2 Wildtype Adult Astrocytoma with WHO Grade 2/3 Histological Features: Molecular Re-Classification, Prognostic Factors, Clinical Outcomes

**DOI:** 10.3390/biomedicines12040901

**Published:** 2024-04-18

**Authors:** Meetakshi Gupta, Mustafa Anjari, Sebastian Brandner, Naomi Fersht, Elena Wilson, Steffi Thust, Michael Kosmin

**Affiliations:** 1Department of Radiotherapy, Guy’s and St Thomas’ NHS Foundation Trust, Great Maze Pond, London SE1 9RT, UK; meetakshi.gupta@nhs.net; 2Department of Radiology, Royal Free Hospital, Royal Free London NHS Foundation Trust, Pond Street, London NW3 2QG, UK; mustafa.anjari@nhs.net; 3Lysholm Department of Neuroradiology, National Hospital for Neurology and Neurosurgery, University College London Hospitals NHS Foundation Trust, Queen Square, London WC1N 3BG, UK; 4Department of Brain Rehabilitation and Repair, Institute of Neurology, University College London, Queen Square, London WC1N 3BG, UK; s.thust@ucl.ac.uk; 5Division of Neuropathology, University College London Hospitals NHS Foundation Trust, Queen Square, London WC1N 3BG, UK; 6Department of Neurodegenerative Diseases, Queen Square Institute of Neurology, University College London, London WC1N 3BG, UK; 7Department of Radiotherapy, University College London NHS Foundation Trust, 250 Euston Rd, London NW1 2PG, UK; 8Sir Peter Mansfield Imaging Centre, School of Physics and Astronomy, University of Nottingham, Nottingham NG7 2RD, UK; 9Queens Medical Centre, Nottingham University NHS Trust, Nottingham NG7 2UH, UK; 10NIHR Biomedical Research Centre, University College London Hospitals NHS Foundation Trust, London W1T 7DN, UK

**Keywords:** IDH-wildtype astrocytoma, early glioblastoma, magnetic resonance imaging, molecular classification

## Abstract

Background: Isocitrate Dehydrogenase 1/2 (IDH 1/2)-wildtype (WT) astrocytomas constitute a heterogeneous group of tumors and have undergone a series of diagnostic reclassifications over time. This study aimed to investigate molecular markers, clinical, imaging, and treatment factors predictive of outcomes in WHO grade 2/3 IDH-WT astrocytomas (‘early glioblastoma’). Methodology: Patients with WHO grade 2/3 IDH-WT astrocytomas were identified from the hospital archives. They were cross-referenced with the electronic medical records systems, including neuroimaging. The expert neuro-pathology team retrieved data on molecular markers—MGMT, *TERT*, IDH, and *EGFR*. Tumors with a *TERT* mutation and/or *EGFR* amplification were reclassified as glioblastoma. Results: Fifty-four patients were identified. Sixty-three percent of the patients could be conclusively reclassified as glioblastoma based on either *TERT* mutation, *EGFR* amplification, or both. On imaging, 65% showed gadolinium enhancement on MRI. Thirty-nine patients (72%) received long-course radiotherapy, of whom 64% received concurrent chemotherapy. The median follow-up of the group was 16 months (range: 2–90), and the median overall survival (OS) was 17.3 months. The 2-year OS of the whole cohort was 31%. On univariate analysis, older age, worse performance status (PS), and presence versus absence of contrast enhancement on diagnostic MRI were statistically significant for poorer OS. Conclusion: IDH-WT WHO grade 2/3 astrocytomas are a heterogeneous group of tumors with poor clinical outcomes. The majority can be reclassified as glioblastoma, based on current WHO classification criteria, but further understanding of the underlying biology of these tumors and the discovery of novel targeted agents are needed for better outcomes.

## 1. Introduction

Adult-type diffuse gliomas are the most common malignant tumors of the central nervous system. Survival varies greatly depending on the subtype of glioma present, with low-grade gliomas having 5-year survival rates as high as 80%, while high-grade gliomas have 5-year survival rates under 5% [1] Regardless of grade and prognosis, gliomas are highly infiltrative and resistant to therapy, rendering them largely incurable. Historically, classification and prognostication for these tumors have been largely based on morphologic features. In the fourth edition of the WHO classification, signs of anaplasia and mitotic activity were used to distinguish WHO grade 2 and 3 gliomas, while the diagnosis of WHO grade 4 glioblastomas required the presence of necrosis and neo-angiogenesis [2]. Recent advances have led to the identification of several molecular subtypes associated with biology and prognosis and have shown that outcomes rely more on the distribution of these molecular subgroups than on biological differences between WHO grade 2 and 3 gliomas per se [3]. The latest WHO 2021 iteration [4] builds on the combination of histological (WHO 2–4) and molecular grading by incorporating additional genetic data as outlined in the cIMPACT-NOW criteria [5]. Consequently, the combination of telomerase reverse transcriptase (*TERT*) promoter mutation, epidermal growth factor receptor (*EGFR*) amplification, and/or copy number changes (+7/−10) in isocitrate dehydrogenase (IDH)-wildtype diffuse astrocytomas defines glioblastoma, IDH-wildtype CNS WHO grade 4, even in cases that microscopically resemble a lower grade [4].

The classification under the 2021 update is dependent largely on isocitrate dehydrogenase (*IDH1/2*) mutation status and 1p/19q codeletion status, resulting in three primary disease groups: IDH-mutant, 1p/19q co-deleted oligodendroglioma; IDH-mutant, non-co-deleted astrocytoma; and IDH-wildtype glioblastoma (GBM). The absence of an IDH mutation is a key characteristic of primary glioblastoma (CNS WHO grade 4), and histological WHO grade 2/3 IDH-wildtype (WT) astrocytomas of the same molecular signature are deemed to represent early glioblastoma stages that can share histological appearances with IDH-mutant astrocytomas [5,6]. IDH-wildtype neoplasms with molecular features of glioblastoma consistently have poorer outcomes compared to IDH-mutant gliomas of the same histological grade [3], however, limited group-specific data exist to inform treatment standards. To what extent this classification will allow for better therapeutic selection and improved clinical outcomes is yet to be seen.

There is no doubt that this cohort represents an aggressive subgroup of glioma that needs intensive treatment, but treatment protocols have varied between centers over the years and were often guided by histological grade and imaging features alone [7]. Standard therapy for GBM encompasses surgical resection followed by chemoradiotherapy, using temozolomide (TMZ) [8]. However, 5-year survival is only 7.2% [9], as almost all GBM tumors locally recur after treatment [10]. Ongoing challenges to GBM treatment include its incomplete resection, high degree of genetic heterogeneity, exclusive blood–brain barrier (BBB), and immunosuppressive microenvironment. Identification of new targeted agents is an important area of research for the treatment of recurrent GBM, as 90% of druggable targets are differentially expressed in a recurrent tumor compared to the tumor at initial diagnosis [11]. Patients may need to receive new therapies throughout their disease. It remains unclear whether the therapeutic advances in glioblastoma can directly be applied to this tumor cohort, and whether the addition of temozolomide produces the same survival benefit. The heterogeneity and plasticity of GBM have limited the success of targeted therapies in clinical trials thus far. Future clinical trials with agents showing strong evidence of anti-tumor activity in relevant pre-clinical models and targeting molecular mutations would likely provide better clinical outcomes. Advances in precision medicine, surgical techniques, and combination therapies will shape the future directions for treatment.

Another issue is the distinction between IDH-WT WHO grade 2 and 3 astrocytomas, which have similar ages at presentation and treatment outcomes (median overall survival (OS) 1.9 years) [12]. The terminology around tumor grading has been simplified, with molecular features dictating classification, and joint histopathologic and molecular analysis determining the grade in the latest WHO update. However, there is a suggestion that grade 2 tumors may behave differently than is evident from the new classification. Magnetic resonance imaging (MRI), in combination with age, has shown predictive and prognostic potential for the WHO grade 2/3 glioma *IDH* genotype [13,14]. Berzero et al. showed that IDH-WT gliomas with grade 2 histology (<2 mitoses per 10 high-power fields), meeting the definition for molecular glioblastoma, had substantially longer OS compared with IDH-WT grade 3 gliomas meeting the definition for molecular glioblastoma (median OS: 57 mo. vs. 17 mo., *p* < 0.0001). They found that most patients with IDH-WT grade 2 gliomas met the cIMPACT criteria because of isolated *TERT* promoter mutations that were not predictive of poor outcomes, hence, some caution is needed when assimilating IDH-WT grade 2 gliomas to molecular glioblastomas, especially those with isolated *TERT* promoter mutation [15]. A recent metanalysis showed that patients with molecular GBM (presence of *TERT* promoter mutation, *EGFR* amplification, or chromosome seven gain and ten loss aberrations) had significantly longer OS times when compared to histological GBM (pooled hazard ratio 0.824, [CI: 0.694–0.98], *p* = 0.03)) and histological grade, age, and surgical extent were significant prognostic factors, with grade 2 molecular GBM showing better OS rates when compared to histological GBM [16].

Imaging features, including gadolinium enhancement patterns, may potentially reflect a prognosis in IDH-WT gliomas [17] with an association with tumor biological evolution. Traditionally, anatomic imaging sequences (T1-weighted, T2-weighted; T1w, T2w) have been used to differentiate high-grade from low-grade glioma or to identify defined genomic aberrations like 1p/19q codeletion in oligodendroglial tumors [18]. However, with advancements in MRI imaging, significant overlap is seen in imaging characteristics, such as the presence of contrast enhancement in low-grade gliomas between WHO grades and genotypes [19]. The use of several physiological imaging sequences, better capturing underlying tumor biology, has been explored. Of these, perfusion imaging has perhaps been studied most extensively for its use in MRI-based glioma grading. Increased contrast enhancement has been associated with shortened progression-free and overall survival in newly diagnosed glioma [20]. There is a huge surge of interest in evaluating the integrative analysis of image features and their association with underlying biology. Machine learning is increasingly being used in fields such as genomics and imaging analysis.

While this expansion in knowledge about these tumors is providing increasing clarity regarding their biological makeup, to what extent their treatment can be intensified and altered, while heavily extrapolating from the treatment protocols of grade 4 tumors, will eventually determine the success of these endeavors. The increasing dependence of classification and prognostication on molecular features increases the importance of laboratory assessment of biomarkers. The WHO update recommends that methylome profiling may also be used as a surrogate marker when a methylome signature is characteristic of an IDH-wildtype glioblastoma in the absence of *IDH* mutation testing—but it cannot serve as a surrogate when targeted therapies and clinical trials require the demonstration of specific mutations before patient treatment [4]. This further highlights the need to determine the clinical implications of these mutations.

We undertook this study to investigate the molecular marker profile of WHO grade 2/3 IDH-WT astrocytomas (‘early glioblastomas’) treated at our institute over a decade and used these markers to re-classify the tumors based on the current WHO update. We also looked at imaging characteristics on the baseline MRI to determine prognostic features, i.e., contrast enhancement on T1w MRI, type of enhancement, pre-surgical tumor volume, and apparent diffusion coefficient (ADC). Another aim of the study was to capture changing trends in treatment protocols over the years and to look for predictive and prognostic factors in this cohort.

## 2. Materials and Methods

Institutional permission from the University College London Research Ethics Committee (UCL REC) was obtained, with informed consent waived for this retrospective data study. Data from 2010 to 2019 containing specific molecular pathology data to identify patients with IDH-WT astrocytomas (WHO grade 2/3) were retrieved from the pathology laboratory information management system and adjunct data resources of the division of neuropathology. Inclusion criteria were patients over 18 years of age with an IDH-WT glioma (grade 2/3), non 1p/19q co-deleted (to exclude oligodendrogliomas), an existing histopathological diagnosis, and a baseline MRI scan. All the tissue samples were analyzed in our neuropathology department at the National Hospital for Neurology and Neurosurgery, University College London Hospitals NHS Foundation Trust, UK. Tumors without evidence of *IDH* 1/2 mutations on immunohistochemistry testing or polymerase chain reaction and multiple Sanger gene sequencing were labeled as IDH-WT. Subsequently, these were cross-referenced with the electronic medical records systems to gather appropriate information about the baseline MRI, treatment details, and follow-up.

The expert neuropathology team retrieved data on molecular markers for all the patients in this study. This included MGMT, *TERT*, *IDH*, and *EGFR*. Additional *TERT* testing was carried out in all cases that were not performed previously. Tumors with *TERT* mutation and/or *EGFR* amplification were reclassified as glioblastoma. Some of these tumors also underwent methylation array profiling, to substantiate the diagnosis.

### 2.1. Imaging Technique and Analysis

All the patients underwent pre-treatment MR imaging studies at 1.5 or 3 Tesla magnetic field strength, including anatomical (T2-weighted, T1-weighted pre- and post-gadolinium) and diffusion-weighted (DWI) sequences. ADC maps were calculated from 3-directional DWI acquired with two gradient values (b = 0 and b = 1000 s/mm^2^). T2-weighted whole lesion volumes and T1-weighted enhancing component volumes were defined by manual segmentation using a dedicated workstation (Olea Sphere, Version 2.3; Olea Medical, La Ciotat, France). Tumor enhancement was categorized into non-enhancing, solid-enhancing, and rim-enhancing, with central necrosis patterns as detailed in prior research [7] Normalized mean ADC values (rADCmean) were calculated by dividing the ADCmean value within the largest solid tumor cross-section by the ADCmean value in normal-appearing centrum semiovale white matter, according to [8]. Tumor location was recorded according to the lesion epicenter (frontal, parietal, paracentral, temporal, insula, occipital, basal ganglia, thalamus, brainstem, or cerebellum), with an additional binary category of ‘deep versus superficial’ established for statistical analysis.

### 2.2. Treatment

All patients underwent surgical intervention after imaging and discussion in our center’s neuro-oncology multidisciplinary meeting, either in the form of maximal safe resection or biopsy. All the patients with IDH-WT tumors were offered post-operative RT with or without concurrent and adjuvant chemotherapy (temozolomide). All patients received 3D planning using CT data with pre-operative and post-operative MRI images. RapidArc (Varian Medical Systems, Inc. Palo Alto, CA, USA) intensity-modulated radiotherapy (IMRT) planning was routinely performed for all patients. In patients with good performance status, long-course radical radiotherapy was commonly delivered to a dose of ≥54 Gy in ≥30 daily fractions over 6 weeks. Alternative short-course radiation of 30 Gy in 6 fractions on alternate days over 2 weeks was used for older patients or patients with limited performance status.

The decision for concurrent temozolomide (TMZ) was taken on an individual basis under consideration of the patient’s performance status and presence of disease high-risk features, consistent with standard-of-care treatment protocols at the time of diagnosis. The Stupp regimen was followed for administering concurrent and adjuvant TMZ [8].

### 2.3. Statistics

SPSS Statistics for Windows, (Version 23.0. IBM Corp.: Armonk, NY, USA) was used for data analysis. Kaplan–Meier plots were generated, and log-rank testing was performed for group comparisons. Cox regression analysis was used for multivariate analysis (MVA). A *p*-value of </= 0.05 was considered statistically significant. Follow-up was calculated from the day of the last radiotherapy fraction.

## 3. Results

### 3.1. Patient Characteristics

Fifty-four patients were identified as eligible for this analysis. The median age of the group was 57 years (range: 20–77). The male-to-female ratio was 1.7:1. Eleven percent (n = 6/54) were ECOG performance status (PS) 0; 65% (n = 35/54) were PS 1; 24% (n = 13/54) were PS 2. 

### 3.2. Tissue Results

Sixty-eight percent (n = 37/54) were high-grade (grade 3) astrocytomas on histopathology. Methylguanine methyltransferase (MGMT)-promoter methylation status was available on all but one patient of whom 60% (n = 32/53) were unmethylated, 5% (n = 3/53) were hypermethylated, and 35% (n = 18/53) had low to moderate levels of methylation. *EGFR* amplification results were available for all the patients. Thirty-nine percent (n = 21/54) were *EGFR*-amplified; sixty-one percent (n = 33/54) did not have *EGFR* amplification. *TERT* testing was carried out retrospectively. It failed in 18 patients despite sufficient samples, and it could not be carried out in another 15 patients due to insufficient tissue or DNA availability. Of the 21 who were successfully tested, 17 (81%) were mutated. Sixty-three percent of the patients (n = 35/54) could be conclusively reclassified as glioblastoma based on either *TERT* mutation or *EGFR* amplification or both (n = 4/35). Additionally, three patients were diagnosed with glioblastoma on a methylation array, of whom one had failed *TERT* testing and two were non-*TERT*-mutated. All three had non-amplified *EGFR*.

### 3.3. Imaging Characteristics

All the lesions were unicentric and most (85% (n = 46/54)) were lobar in the epicenter (n = 17 temporal lobe, n = 15 frontal lobe, n = 11 parietal lobe, n = 2 insula, and n = 1 paracentral lobule). A smaller number of tumors were deep-seated (thalamus n = 5, brainstem n = 2, and basal ganglia n = 1). Examples of the tumor locations are provided in Figure 1.

Sixty-five percent (n = 35/54) of the masses showed contrast enhancement, of which 18 showed a solid pattern of enhancement, and 17 showed rim enhancement with central necrosis. The median (±SD) volume of the T1-weighted enhancing tumor component was 0.45 cc (±8.16). The median (±SD) T2-weighted entire tumor volume measured 66.53 cc (±117.55). The largest cross-section solid tumor median (±SD) rADCmean value was 1.45 (±0.3). Examples of glioma contrast enhancement patterns and ADC values are shown in Figure 2.

### 3.4. Treatment Characteristics

Seventy percent (n = 38/54) of the patients underwent only a biopsy for tissue diagnosis, 11% (n = 6/54) underwent debulking surgery, and 19% (n = 10/54) underwent complete macroscopic resection. The median time from diagnostic MRI to surgical intervention was 8 days (IQR 4–36 days). Seventy-two percent (n = 39/54) received long-course radical radiotherapy, of whom 64% (n = 25/39) also received concurrent TMZ. The use of concurrent TMZ altered during the period under analysis. Of the twenty-one patients receiving high-dose radiotherapy (≥54 Gy total dose) before 2015, only seven (33%) received concurrent TMZ chemotherapy. All the patients receiving high-dose radiotherapy (n = 18) after January 2015 received concurrent TMZ.

Two patients received a three-week course of RT (40 Gy in 15 daily fractions) with concurrent TMZ, and thirteen received short-course RT (30 Gy in 6 fractions on alternate days over two weeks) only. Of the 25 patients receiving concurrent chemotherapy, 19 also received adjuvant TMZ. This was omitted in five patients who had grade >3 myelosuppression on concurrent TMZ therapy, and one who was treated within the CATNON trial (NCT00626990). Table 1 shows a summary of patient characteristics.

### 3.5. Survival Analysis

The median follow-up of the group was 16 months (range: 2–90). Eleven patients were alive at the time of analysis. The median follow-up of patients who were alive at the time of analysis (n = 11/54) was 26 months (range: 7–90). The median overall survival (OS) was 17.3 months (Figure 3a). The 2-year OS of the whole cohort was 31%. On univariate analysis (UVA), age at diagnosis >57 years (dichotomized at the median) [median (95% CI): 20 (12.5 to 27.5) vs. 14 (9.8 to 18.8) months; *p* = 0.013], worse PS [0 vs. 1 vs. 2, median (95% CI): 24 (0 to 60) vs. 18.3 (14 to 22.6) vs. 14.3 (0.7 to 27.8) months; *p* = 0.004] and presence versus absence of contrast enhancement on diagnostic MRI [median (95% CI): 15.2 (12.1 to 18.3) vs. 28.8 (21.3 to 36.2) months; *p* = 0.003] were statistically significant for poorer OS (Figure 3b). Table 2 shows the detailed results of the univariate analysis. No difference in survival was evident based on the location of the lesion on MRI (lobar vs. thalamic, brainstem, or basal ganglia), the pattern of gadolinium enhancement of the lesions (solid/patchy versus rim-enhancing with central necrosis), pre-operative tumor volume, or the proportion of enhancement within the tumor. Sex, type of surgery (biopsy vs. resection; complete resection vs. incomplete resection), and time to surgery were not predictive of OS via UVA in this cohort. The dose of radiation in the radical group [high dose vs. low dose, median (95% CI): 18.2 (14.2 to 22.16) vs. 16 (12.7 to 19.19) months; *p* = 0.195], the addition of chemotherapy [median (95% CI): 19.8 (15.7 to 23.9) vs. 16 (11.9 to 19.9) months; *p* = 0.39], and MGMT status [methylated vs. unmethylated, median (95% CI): 16.67 (13.2 to 20.14) vs. 18.3 (12 to 24.65) months; *p* = 0.40] were not predictive of OS. The OS of patients treated with palliative short-course RT did not differ significantly from those treated radically [radical vs. palliative, median (95% CI): 18.2 (14.2 to 22.16) vs. 16 (12.7 to 19.19) months; *p* = 0.195]. On multivariate analysis, none of these factors were significant predictors of OS.

## 4. Discussion

This study aimed to identify treatment outcomes and prognostic factors for IDH-WT histological grade 2/3 astrocytomas with molecular features of glioblastoma, IDH-wildtype, CNS WHO grade 4, and to capture changing treatment trends. Although the included patients were treated heterogeneously based on the understanding of the disease and evidence available at the time, the treatment regimen generally consists of a standard regimen of RT plus concurrent and adjuvant temozolomide, based on extrapolation from patients with glioblastoma.

The recently published 5th Edition of the WHO Classification of Tumors of the Nervous System now mandates that IDH-wildtype diffuse and astrocytic glioma in adults are diagnosed as glioblastoma, even in the absence of histological high-grade features (microvascular proliferation or necrosis), when there is a *TERT* promoter mutation and/or *EGFR* gene amplification and/or chromosome 7 gain/chromosome 10 loss [4]. We could reclassify around 60% of these patients as IDH-WT glioblastoma based on the current WHO classification. Only three patients who had both *EGFR* and *TERT* data did not have the molecular characteristics of glioblastoma.

We did not observe an improvement in OS in our cohort of patients either with the use of high-dose radiotherapy or the addition of concurrent (and adjuvant) TMZ, irrespective of the MGMT molecular status. We separately analyzed patients who were reclassified as glioblastoma, and there was still no benefit with the addition of chemotherapy regardless of MGMT status. Although toxicities of systemic therapy have not been reported in the current series, we did observe a 20% incidence of grade 3/4 hematological toxicity, warranting discontinuation of adjuvant chemotherapy. Advanced age and limited PS were indicative of poorer outcomes, like glioblastoma. The timing and type of surgical intervention did not influence outcomes, which remained poor throughout the cohort. Although the study failed to show a survival benefit with the addition of chemotherapy, high doses of radiotherapy, and extensive surgery, the sample size was too small to conclusively determine the effect of these interventions.

Our findings align well with the published results of the CATNON trial, which showed that the addition of TMZ to IDH-WT tumors did not result in improved overall survival either in the concurrent or adjuvant setting [21]. Research into identifying the underlying differences between this group and WHO grade 4 IDH-wildtype glioblastoma, including potentially targetable mutations, may assist in the identification of new therapeutic avenues. Further subgrouping of this cohort based on additional molecular testing is the subject of ongoing research. The treatment paradigms for this disease will continue to evolve as our understanding of its biology and behavior advances, with a particular challenge being to define optimal therapy for such tumors that are not reclassified as glioblastoma.

The benefit obtained by the extent of surgical resection and high-dose radiation also remains uncertain in these patients. What we do know, however, is that the outcomes of this WHO grade 2/3 IDH-WT cohort with surgery and radiation are similar to the glioblastoma group and worse than IDH-mutant astrocytomas, irrespective of WHO grade. Treating tumors as unique entities and exploring tailored treatment options through well-designed clinical trials is the only way to improve survival in this group.

Furthermore, we explored diagnostic MRI-based prognostication in our cohort. According to previous research, the presence of contrast enhancement may predict shorter survival [22], which could be explained by pathological neovascularization and blood–brain barrier breakdown as a characteristic of more advanced disease. Unlike previous reports [23], we did not observe an association between lesion volume and outcome in this group. An inverse correlation between glioblastoma ADC values and survival has been suggested [24,25]; however, results are inconsistent between studies, with some identifying no such association [21]. The only parameters that appeared to affect overall survival from this study were age, PS, and contrast enhancement on pre-operative MRI. These are established prognostic factors for gliomas and hence continue to guide the management of these tumors. While age and PS essentially determine to what extent the patient can undergo more aggressive debulking and receive higher doses of radiotherapy and chemotherapy, contrast-enhancing lesions on MRI consistently contain the highest density of tumor cells along with the most aggressive histological features in malignant glioma and hence represent a group of tumors with poorer prognosis [21,26]. The study indicates the need for ongoing research to refine the management of this unique tumor entity and that further studies would establish prognostic features to steer treatment guidelines.

The strength of our study is in the inclusion of a large database of patients representing a comparatively rare histo-molecular group of gliomas treated at a single center, and the availability of all the tumor samples at the neuropathology facility, which were used to systematically test the molecular markers (which were not historically available). The emphasis on molecular classification of brain tumors in WHO updates has emphasized the need for state of art pathology services and expertise to satisfactorily interpret the results and guide management. We were able to confirm the reclassification to glioblastoma in a large part of our cohort while also establishing the challenge our expert team faced in retrospectively testing *TERT* mutation in the preserved samples. Another advantage of our study is the availability of systematic MR imaging for all the patients, which allowed for the correlation of MR parameters with outcomes and exploration of prognostic features that could potentially guide therapy in the future.

A limitation of this work is the retrospective nature of the analysis, alongside the changing nature of our understanding of the disease entity and its treatment during the timeframe under evaluation. Furthermore, in the multivariate analysis, none of the parameters resulted significantly, which was probably due to the smaller sample size. However, considering the rarity of this tumor and recent changes in classification, which makes extrapolation from older studies difficult, this study provides fresh insight into the ability to re-classify tumors in real-time from archived samples at institutes with the capacity and agency to do so and be able to pool data from several large centers worldwide to steer standard guideline development to improve outcomes in this cohort. Our team successfully retrieved all the molecular information from patient records and retrieved samples to perform tests that were not standard of care at the time of original diagnosis, which could encourage other institutes with large patient databases to undertake similar work.

## 5. Conclusions

The recent WHO update reclassifies all IDH-WT gliomas into glioblastomas based on molecular markers (presence of *TERT* promoter mutation, *EGFR* amplification, or chromosome seven gain and ten loss aberrations). While this is a direct result of the evolving understanding of the biology of these tumors, more needs to be done to understand the prognosis of smaller subgroups, grades 2 vs. 3, cohorts with isolated *TERT* mutations. Varied treatment pathways have been adopted for patients with IDH-WT gliomas over the last decade, many of which are directly extrapolated from glioblastoma. With the recent trials showing little or no benefit with this approach, management innovation tailored specifically for this cohort will be imperative for better treatment outcomes. MRI remains an important and powerful tool for understanding tumor biology, and a comprehensive approach needs to be taken to further improve the histo-molecular classification.

## Figures and Tables

**Figure 1 biomedicines-12-00901-f001:**
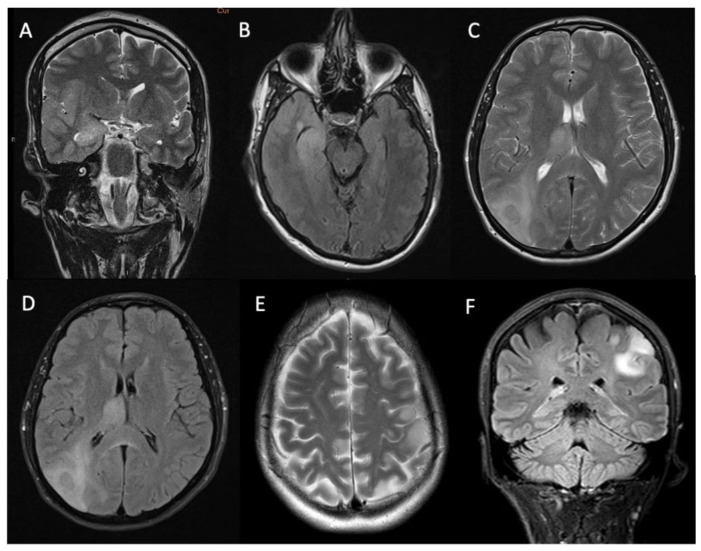
T2-weighted and FLAIR images showing glioma locations in the right temporal lobe (patient 1—**A**,**B**), right parietal lobe (patient 2—**C**,**D**) with infiltration into the deep white, matter and thalamus, and in the left paracentral lobule (patient 3—**E**,**F**).

**Figure 2 biomedicines-12-00901-f002:**
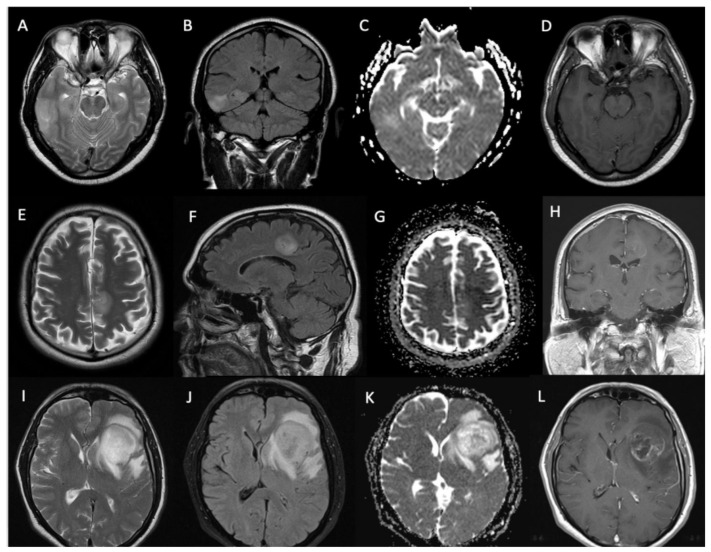
T2-weighted and FLAIR images, ADC maps and contrast-enhanced T1-weighted images in 3 different patients showing gadolinium enhancement patterns with no contrast uptake (patient 1, **A**–**D**), patchy-solid contrast uptake (patient 2, **E**–**H**) and limited rim-enhancement surrounding central necrosis (patient 3, **I**–**L**). Of note, the rim-enhancement in L is less pronounced than in typical WHO grade 4 glioblastoma.

**Figure 3 biomedicines-12-00901-f003:**
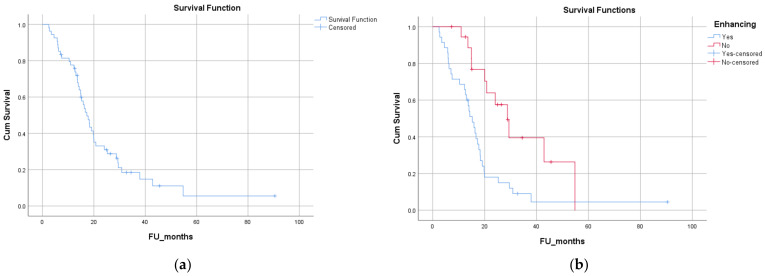
Kaplan–Meier curves for overall survival (OS) of the entire cohort (**a**) and effect of MRI contrast enhancement on OS (**b**): blue = presence of contrast enhancement; red = absence of contrast enhancement.

**Table 1 biomedicines-12-00901-t001:** Summary of tissue, radiological, and treatment characteristics.

Characteristic	Numbers (Percentage)
Tumor grade	
Grade 2	17 (32%)
Grade 3	37 (68%)
MGMT promoter methylation	
Unmethylated	32 (60%)
Low to moderate methylation	18 (35%)
Hypermethylated	3 (5%)
*EGFR* amplification (n = 54)	
Amplified	21 (39%)
Unamplified	33 (61%)
*TERT* mutation (n = 21)	
Mutated	17 (81%)
Non-mutated	4 (19%)
Radiological characteristics	Numbers (percentage)
Location (n = 54)	
Lobar	46 (85%)
Deep-seated	8 (15%)
Contrast enhancement	
Solid pattern	18 (33%)
Rim enhancement	17 (31%)
Non-enhancing	19 (36%)
Treatment characteristics	Numbers (percentage)
Surgical procedure	
Biopsy	38 (67%)
Debulking	6 (11%)
Complete excision	10 (19%)
Radiotherapy dose	
≥54 Gy (1.8–2 Gy/fraction)	39 (72%)
40 Gy (2.67 Gy/fraction)	2 (4%)
36 Gy (6 Gy/fraction)	13 (24%)
Chemotherapy	
Concurrent	25 (46%)
Concurrent and adjuvant	19 (35%)
None	10 (19%)

**Table 2 biomedicines-12-00901-t002:** Results of univariate analysis.

Test Variable	Overall Survival [Median (CI)]	*p*-Value
Age at diagnosis (dichotomized at median—≤57 years vs. >57)	20 (12.5 to 27.5) vs. 14 (9.8 to 18.8) months	*p* = 0.013
Sex (female vs. male)	17.3 (14.5 to 20) vs. 17.8 (12.3 to 23.2) months	*p* = 0.36
Performance status (0 vs. 1 vs. 2)	24 (0 to 60) vs. 18.3 (14 to 22.6) vs. 14.3 (0.7 to 27.8) months	*p* = 0.004
Contrast enhancement on diagnostic MRI (present vs. absent)	15.2 (12.1 to 18.3) vs. 28.8 (21.3 to 36.2) months	*p* = 0.003
Location of the lesion on MRI (superficial vs. deep)	18.3 (15.3 to 21.2) vs. 14.9 (12.7 to 17) months	*p* = 0.32
Pattern of gadolinium enhancement (solid/patchy versus rim-enhancing with central necrosis)	15.9 (12.5 to 19.3) vs. 14 (11.7 to 16.2) months	*p* = 0.57
Pre-operative tumor volume on MRI (dichotomized at median—66.5 cc)	20 (15.6 to 24.2) vs. 15 (12.3 to 17.6) months	*p* = 0.20
ADC value (dichotomized at median—1.45)	15.2 (11.4 to 19) vs. 19.8 (17.3 to 22.3) months	*p* = 0.43
Type of surgery (biopsy vs. maximal safe resection)	17.8 (15.5 to 20) vs. 14 (11.3 to 16.6) months	*p* = 0.91
Complete resection vs. incomplete resection	14.9 (7 to 22.9) vs. 17.3 (14.5 to 20) months	*p* = 0.28
Time to surgery (from diagnostic MRI—<1 week vs. >1 week)	15.2 (12.2 vs. 18.2) 18.3 (15 to 21.5)	*p* = 0.72
Radical vs. palliative radiotherapy	18.2 (14.2 to 22.16) vs. 16 (12.7 to 19.19) months	*p* = 0.20
Dose of radiation (high dose vs. low dose)	18.2 (14.2 to 22.16) vs. 16 (12.7 to 19.19) months	*p* = 0.20
Concurrent chemotherapy	19.8 (15.7 to 23.9) vs. 16 (11.9 to 19.9) months	*p* = 0.39
MGMT status (methylated vs. unmethylated)	16.67 (13.2 to 20.14) vs. 18.3 (12 to 24.65) months	*p* = 0.40

## Data Availability

The data will be made available upon reasonable request.

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
