# Peer review of "Isocitrate Dehydrogenase 1/2 Wildtype Adult Astrocytoma with WHO Grade 2/3 Histological Features: Molecular Re-Classification, Prognostic Factors, Clinical Outcomes"

_biomedicines, 2024, doi:10.3390/biomedicines12040901_

Round 1
Reviewer 1 Report
Comments and Suggestions for Authors
This study showed the changes of diagnostic criteria affect clinical diagnosis. Now we can diagnose these tumors using new diagnostic criteria. Author did not fully explain the meaning of this study.
Author Response
Many thanks for reviewing the manuscript and the comments.
Reviewer 1: Author did not fully explain the meaning of this study
Response: We have redone parts of the introduction and discussion after reflecting on this
Reviewer 2 Report
Comments and Suggestions for Authors
In this paper Gupta et al present the results of a retrospective monocentric analyses of IDH1/2-WT astrocytoma WHO 2/3. They included 54 patients and re-classified 63% as glioblastoma based on novel WHO classification. They conclude that older age, worse PS and presence of contrast enhancement on MRI were significant for poorer OS.
The paper is well written and easy to understand. Tables and figures are essential to comprehension but more are needed. Autocitation rate is low (1/17). It fits the Special Issue Topic.
However, several methodological issue need to be addressed in order to improve the overall quality:
-it must be stressed that at the multivariate analysis none of the parameters resulted significant. This probably due to the little sample.
- conclusions regarding surgery and radio/chemiotherapy have little statistical evidence. Data about localisation, pre-op volume, post-op volume, concurrent multiple localisation are lacking. Therapy regimens are variable and thus the sample has a wide internal variability. To infere that surgery and chemio/radiotherapy have no impact on OS is, at least, excessive.
- it is not clear if the multicentric lesions subgroup has a significant different OS, but it could be interesting.
Author Response
Many thanks for reviewing the manuscript and for the comments
Reviewer 2:
-it must be stressed that at the multivariate analysis none of the parameters resulted significant. This probably due to the little sample.
Response: This has been clarified in the discussion
- conclusions regarding surgery and radio/chemiotherapy have little statistical evidence. To infere that surgery and chemio/radiotherapy have no impact on OS is, at least, excessive.
Response: This has been clarified in the discussion
-Data about localisation, pre-op volume, post-op volume, concurrent multiple localisation are lacking. Therapy regimens are variable and thus the sample has a wide internal variability.
Data regarding localisation is provided in detail including the results ion UVA. There were no multiple lesions. No difference in survival was evident based on pre-operative tumor volume or the proportion of enhancement within the tumor. This has been clarified in the manuscript
- it is not clear if the multicentric lesions subgroup has a significant different OS, but it could be interesting.
Response: all lesions were unicentric and this has been included in the manuscript to clarify.
Reviewer 3 Report
Comments and Suggestions for Authors
1. I agree with the highlighted sections and the Importance of the study sections but they should be removed as they do not adhere to MDPI guidelines.
2. Please include statistics at the beginning of the Introduction section regarding the prevalence or potential mortality rate of the tumor discussed in this study. This addition will underscore the significance and necessity of our research project.
3. The literature review in the Introduction section is inadequate and requires improvement. Classify previous papers in this field and identify the research gap that you aim to address.
4. The hypothesis presented in the final paragraph of the Introduction is unclear to readers and needs clarification.
5. In the statement "Institutional permission was obtained with informed consent waived for this retrospective data study," please specify the name of the IRB center. Also, consider adding "...analyzed in our neuropathology department," along with the name of our university/hospital.
6. Please include the word "adult" in the paper's title.
7. The sample size is too small to draw meaningful conclusions. Provide p-values for all parameters in Table 1 to understand why these results are not reliable.
8. Reproduce Figure 3, as it is of low quality. Additionally, the follow-up time of 100 months, and the minimum is 2 months? The standard follow-up time is at least 6 months. You can not remove the patients with a follow-up time of less than 6 months, because it makes bias in findings. Consider contacting participants to update their status.
9. The primary finding pertains to the Kaplan-Meier curve concerning the presence or absence of contrast enhancement. Provide a clinical interpretation of these findings in the Discussion section, as it currently lacks justification for readers.
10. Especially considering comments 7 and 8, I am not optimistic about accepting this paper. However, I am open to allowing the authors an opportunity to respond to these concerns. In addition, please work on improving the English language level of the paper.
Comments on the Quality of English LanguageModerate
Author Response
Many thanks for reviewing the manuscript and for the comments
Reviewer 3:
1. I agree with the highlighted sections and the Importance of the study sections but they should be removed as they do not adhere to MDPI guidelines.
Response: These have been removed
2. Please include statistics at the beginning of the Introduction section regarding the prevalence or potential mortality rate of the tumor discussed in this study. This addition will underscore the significance and necessity of our research project.
Response: This has been added to the introduction
3. The literature review in the Introduction section is inadequate and requires improvement. Classify previous papers in this field and identify the research gap that you aim to address.
Response: This has been elaborated in the introduction
4. The hypothesis presented in the final paragraph of the Introduction is unclear to readers and needs clarification.
Response: This has been elaborated in the introduction
5. In the statement "Institutional permission was obtained with informed consent waived for this retrospective data study," please specify the name of the IRB center. Also, consider adding "...analyzed in our neuropathology department," along with the name of our university/hospital.
Response: Both have been amended
6. Please include the word "adult" in the paper's title.
Response: This has been amended
7. The sample size is too small to draw meaningful conclusions. Provide p-values for all parameters in Table 1 to understand why these results are not reliable.
Response: Table 1 contains only demographic characteristics. We have not included any statistical analysis. But this statement to emphasize the small sample size is included in discussion.
8. Reproduce Figure 3, as it is of low quality. Additionally, the follow-up time of 100 months, and the minimum is 2 months? The standard follow-up time is at least 6 months. You can not remove the patients with a follow-up time of less than 6 months, because it makes bias in findings. Consider contacting participants to update their status.
Response: The figure has been reproduced.
Only 11 patients were alive at the time of analysis. The patient with a 2-month follow-up had died within 2 months of treatment completion. We have added follow-up of patients who were alive at the time of analysis.
9. The primary finding pertains to the Kaplan-Meier curve concerning the presence or absence of contrast enhancement. Provide a clinical interpretation of these findings in the Discussion section, as it currently lacks justification for readers.
Response: This has been clarified further
10. Especially considering comments 7 and 8, I am not optimistic about accepting this paper. However, I am open to allowing the authors an opportunity to respond to these concerns. In addition, please work on improving the English language level of the paper.
Response: A Grammarly check was run through the text and English language changes were implemented
Round 2
Reviewer 1 Report
Comments and Suggestions for Authors
I recognized significant improvements of this manuscript.
Author Response
Many thanks for reviewing the revisions
Reviewer 2 Report
Comments and Suggestions for Authors
No further comments. Nice paper, appreciations for the authors.
Author Response
Many thanks for reviewing the revisions
Reviewer 3 Report
Comments and Suggestions for Authors
Thank you for the corrections. It is necessary to compute the p-value for all measured parameters and report them in the paper to determine which ones are meaningful. Consequently, you can discuss the findings based on the meaningful parameters. Currently, we cannot make a decision about the results and discussion, especially regarding the findings discussed.
Author Response
Many thanks for reviewing the revisions. We have added a second table with the detailed results of the univariate analysis and p values.